# The Neuroprotective Effect of Tea Polyphenols on the Regulation of Intestinal Flora

**DOI:** 10.3390/molecules26123692

**Published:** 2021-06-17

**Authors:** Zhicheng Zhang, Yuting Zhang, Junmin Li, Chengxin Fu, Xin Zhang

**Affiliations:** 1Key Laboratory of Conservation Biology for Endangered Wildlife of the Ministry of Education, College of Life Sciences, Zhejiang University, Hangzhou 310058, China; qiushizj@tzc.edu.cn; 2Taizhou Biomedical Industry Research Institute Co., Ltd., Taizhou 317000, China; 3College of Life Sciences, Taizhou University, Taizhou 317000, China; 4Department of Food Science and Engineering, Ningbo University, Ningbo 315211, China; yutingzhang132@163.com

**Keywords:** tea polyphenols, neuroprotective effect, intestinal flora, microbe-gut-brain axis

## Abstract

Tea polyphenols (TPs) are the general compounds of natural polyhydroxyphenols extracted in tea. Although a large number of studies have shown that TPs have obvious neuroprotective and neuro repair effects, they are limited due to the low bioavailability in vivo. However, TPs can act indirectly on the central nervous system by affecting the “microflora–gut–brain axis”, in which the microbiota and its composition represent a factor that determines brain health. Bidirectional communication between the intestinal microflora and the brain (microbe–gut–brain axis) occurs through a variety of pathways, including the vagus nerve, immune system, neuroendocrine pathways, and bacteria-derived metabolites. This axis has been shown to influence neurotransmission and behavior, which is usually associated with neuropsychiatric disorders. In this review, we discuss that TPs and their metabolites may provide benefits by restoring the imbalance of intestinal microbiota and that TPs are metabolized by intestinal flora, to provide a new idea for TPs to play a neuroprotective role by regulating intestinal flora.

## 1. Introduction

The human intestinal tract contains as many as 100 trillion microorganisms, which are composed of at least 1800 genera and as many as 40,000 bacteria, containing 100 times as many genes as humans [1]. These complex microbial communities and microflora undergo a vigorous development process throughout the whole life cycle and establish a symbiotic and harmonious rapport with the host early in life [2]. In general, the two most dominant genera in the intestinal microbiota are *Bacteroides* and *Firmicutes*, which contain gram-negative bacteria and gram-positive bacteria respectively [3]. It is now clear that intestinal microflora plays a key role in the health of the host, defending against pathogens, metabolizing nutrients and drugs in the diet, and affecting the absorption and distribution of fats in the diet [4]. However, the influence of microflora extends even beyond the gastrointestinal tract, as the overall balance of its composition, coupled with the key species that elicit specific responses, can impact the central nervous system and regulate brain function through the brain–gut axis [5]. Besides, the early life interference of the intestinal microflora in a developing body may also affect neurodevelopment [3].

Tea (*Camellia sinensis* (L.) O. Kuntze) has been used as a beverage for more than 2000 years. It contains a variety of chemicals, such as polyphenols, methylxanthine, caffeine, lipids, amino acids, minerals, and volatile compounds. The main active components in tea extract are tea polyphenols (TPs), which are the compounds of many kinds of phenolic substances [6]. Flavonoids are the main components of TPs, including catechins and their derivatives. Catechins mainly include epigallocatechin gallate (EGCG), epigallocatechin (EGC), epicatechin gallate (ECG), and epicatechin (EC), of which EGCG is the most abundant [7]. The molecular structure of the major TPs is shown in Figure 1. In recent years, TPs, as a kind of neuroprotective agent, have attracted extensive attention. Because of their anti-oxidation, scavenging free radicals, chelating metal ions, anti-cancer, anti-inflammation, and anti-apoptosis, TPs are potential bioactive compounds with neuroprotection and neuromodulation [8]. Moreover, the host can get the TPs into the brain tissue to play their neuroprotective role simply by drinking tea. For example, after direct oral administration of labeled EGCG in mice, it was found that EGCG was distributed in many organs of the body, including the brain [9]. Although a small number of studies have reported the intake and brain distribution of edible TPs, problems concerning stability, bioavailability, and metabolic transformation of TPs remain to be further solved. What is more, the physiological functions of TPs are closely related to bioavailability [10]. Intestinal flora can biotransform TPs to form simpler and easily absorbable phenolic substances into the human circulatory system, which not only increases the concentration of phenolic compounds in the body, but also improves the biological activity of the parent compounds [11].

## 2. The Relationship between Gut Microbiota and Nervous System

The brain–gut axis is a communication system that integrates nerve, hormone, and immune signals between the brain and the intestinal tract [18]. Intestinal microbiota and their metabolites can participate in the regulation of nervous system function through the above pathways and consequently affect the pathogenesis of nervous system-related diseases. Likewise, the nervous system can affect the composition and number of intestinal microorganisms through the above pathways [19]. In recent years, studies found that intestinal flora can communicate with the brain through the brain–gut axis, regulate intestinal function and brain development, and change host behavior (Figure 2). Moreover, these studies explain the pathogenesis of many diseases, especially neurological abnormalities, which provides a new direction for its treatment and research [20]. Although some studies have focused on exploring bi-directional communication between the microflora and the nervous system, more is focused on the modification of the nervous system by the intestinal microflora. Here are some of the mechanisms by which the gut and its microflora affect the nerves and behavior of the brain.

### 2.1. Neural Pathways

The regulation of the gastrointestinal nervous system is composed of the central nervous system (CNS), autonomic nervous system, and enteric nervous system (ENS) [21]. The ENS can directly transmit the information of intestinal sensation to the brain through its intestinal nerve and vagal afferent nerve [20]. Intestinal microbes can affect the development and function of the intestinal nerve by activating pattern recognition receptors, especially Toll-like receptors (TLR) such as TLR-2 and TLR-4 [22]. A study in a model of ENS transgenic zebrafish showed that the lack of ENS could transform the GI microflora into pro-inflammatory microflora, while the transplantation of ENS precursors restored the microflora to normal, indicating an interaction between intestinal microflora and ENS [23].

The vagus nerve pathway is the main way for intestinal flora to affect the CNS [24]. In addition, the vagus nerve is the main afferent pathway from the abdominal cavity to the brain. Vagotomy blocks the anti-anxiety effect of *Bifidobacterium* spp. in antihypertensive inhibition and avoidance [25]. In a mouse model of low-grade colitis, *Bifidobacterium longum* normalized the anxiety-like behavior in mice by activating the vagus nerve [26]. Similarly, the evidence that intestinal bacteria use vagus nerve afferent neurons to change the emotional and behavioral response of the host comes from research on the change of c-Fos expression in vagus nerve afferent cytons after oral administration of *Campylobacter jejuni*. Previous studies have indicated that subclinical oral *Campylobacter jejuni* can induce anxiety-related behaviors without triggering a systemic immune response [27]. This induction is associated with the increased expression of the c-Fos gene in bilateral vagal neurons activated by the brainstem, while the expression of the c-Fos gene in the paraventricular hypothalamic nucleus reduced significantly after subphrenic vagotomy in mice. Interestingly, the gene expression of nucleus accumbens was also altered in animals with subdiaphragmatic vagal deafferentation [28]. It was reported that long-term administration of *Lactobacillus rhamnosus* promoted exploratory behavior in mice. It has to do with the specific changes of the γ-aminobutyric acid (GABA) system in the brain, which greatly rests with the integrity of the vagus nerve [29]. Although there is plenty of evidences that the vagus nerve is the signal channel from the microbiota to the brain, further research is needed to unravel the specific mechanism of these pathways.

### 2.2. Immune Pathway

The immune system secretes a variety of cytokines that express several antigenic markers and control neurotransmitters during activation to initiate the neuroinflammatory response [30]. Intriguingly, a recent study suggested that animals treated with germ free (GF) and antibiotics showed defects in the maturation, activation, and differentiation of microglia, eventually resulting in the inadequate immune response to various pathogens [31]. Microglia cells are the most abundant immune cells in the brain, accounting for about 5–20% of central glial cells. They play a vital role in the immune function of the CNS system, including phagocytosis, production of cytokines, and activation of inflammation [32]. Tumor necrosis factor-α produced by activated microglia is the cause of monocyte migration. The migration of monocyte was reversible when treated with eight probiotic combinations, including four *Lactobacillus* spp., three *Bifidobacteria* spp., and salivary *Streptococcus* subspecies, which further confirmed that the intestinal microflora could affect the dynamic balance of the central nervous system [33]. Another study indicated that transgenic mice lacking the lymphocyte recombination activated gene 1 (Rag1) experienced changes related to cognitive and anxiety performance during behavioral tasks. However, the changes could be reversed by combined intervention with *Lactobacillus rhamnose* and *Lactobacillus helveticus* [34]. Therefore, probiotics can potentially mitigate adaptive immune damage and subsequent behavioral changes, indicating the key effect of the immune system in the gut–brain axis. In addition to the well-established immune functions in the CNS, structurally active microglia are involved in neuronal activities at different stages of development and adulthood, including synaptic remodeling to improve neural network signals [35]. Recently, the researchers found that the absence of complex host microflora (when treated with GF animals or mice treated with antibiotics) increased the number of microglia and led to defects in the maturation, activation, and differentiation of microglia, which eventually resulted in impaired immunity to bacterial or viral infections [36]. These findings display healthy and diverse gastrointestinal microflora that is essential for the continued preservation of healthy microglia and normal brain function throughout our lives [37].

Intestinal microorganisms and intestinal mucosal cells are believed to modulate the activation of immune molecules to affect the central nervous system. Presently, several studies have already proved that intestinal microflora can regulate peripheral inflammatory responses, thereby influencing brain function and behavior [38,39]. These immune molecules include pro-inflammatory factors like cytokines IL-1 and IL-6 and anti-inflammatory factors like IL-10 and transforming growth factor β [40]. Additionally, gram-negative bacteria can stimulate the production of pro-inflammatory cytokines like interleukin (IL)-6 and IL-1B by binding to Toll-like receptors (TLR-4) expressed on monocyte macrophages and microglia through the lipopolysaccharide (LPS) components of their cell walls. In cases of irritable bowel syndrome and depression, it is thought that decreased intestinal permeability leads to the transfer of bacteria from the intestinal cavity to the systemic circulation, where they can trigger an inflammatory response by stimulating TLR-4 on circulating immune cells [41]. Whereas Streptococcal infection will induce the production of reactive antibodies in the central nervous system and lead to abnormal behavior. This pathological syndrome is called pediatric autoimmune neuro psychosis concerning streptococcal infection [42].

### 2.3. Neuroendocrine Pathway

Intestinal microbes can influence the central nervous system, leading to behavioral and cognitive changes by affecting the release of a variety of neurotransmitters, including norepinephrine (NE), 5-hydroxytryptamine (5-HT), brain-derived neurotrophic factor (BDNF), and dopamine (DA) [43]. However, not only that, intestinal microflora can also secrete a variety of neurotransmitters, such as GABA, catecholamine, and histamine, and transmit signals to CNS through intestinal chromaffin cells and/or intestinal nerves [44].

5-HT is an important monoamine neurotransmitter catalyzed by the enzyme TPH in intestinal chromaffin cells, which is involved in the regulation of almost all brain functions and many physiological processes, such as mood, sleep, pain, and aggression behavior. Unbalance of the serotonergic system has been connected with the development of many neuropsychiatric disorders, including anxiety and depression [45]. Reduced production of 5-HT, lacking a 5-HT receptor site, or inability of 5-HT to reach the receptor sites are all underlying problems that can result in dysfunction [46]. Moreover, it was recently demonstrated that some metabolites from the *Clostridium perfringens* of mouse and human microbiota, such as single-chain fatty acids (SCFAs), especially acetic acid and butyrate, can stimulate intestinal chromaffin cells to produce serotonin by increasing the expression of TPH1 [47]. Notably, the regulation of intestinal serotonin production by microorganisms has a significant effect on body balance. Some studies have confirmed intestinal flora can cause gastrointestinal discomfort in autistic children through the 5-HT pathway [48]. To sum up, the interaction between microorganisms and the brain–gut axis may be associated with the neuroendocrine pathway, and the specific mechanism needs to be further studied.

### 2.4. Microbial Metabolite Pathway

It has been reported that microorganisms affect the neurophysiological changes of the host by metabolizing chemicals that bind to receptors inside and outside the intestine as well [49]. Short-chain fatty acids (SCFAs) are produced by the fermentation of dietary fiber by intestinal microorganisms, mainly including acetate, propionate, and butyrate [50]. Vioricaetal et al. found that the SCFAs mixture regulated intestinal barrier function by increasing transepithelial resistance and decreasing paracellular permeability. In a rat model of transient focal cerebral ischemia, intraperitoneal injection of sodium butyrate attenuated damage at the blood–brain barrier (BBB) [51]. In animal models, SCFAs improved the neurodevelopment and cognitive function in animals with neurodegenerative diseases. The metabolites of intestinal flora can also enter the blood circulation and act on the whole body, but its potential effect on the host is not clear [52]. Several research groups indicated that 4-ethylbenzene sulfate, a metabolite regulated by microorganisms, was sufficient to induce anxiety-like behavior in mice [22]. The increased permeability of BBB in GF mice was mainly related to the decreased expression of tight junction proteins (such as occludin and claudin-5), which could be alleviated by SCFAs [53]. For example, the expression of occludin in these GF mice increased after oral administration of sodium butyrate [54]. As mentioned earlier, the morphology and maturity of microglia in GF mice also got improvement after SCFAs treatment. Furthermore, SCFAs produced by intestinal microflora can pass through the circulatory system through the BBB, affect microglia, and control their translation [55].

### 2.5. Nitric Oxide Pathway

Nitric oxide (NO) acts as the principal neurotransmitter of the nonadrenergic noncholinergic ENS and performs a vital role in the CNS of humans and animals [56]. However, excessive NO production becomes noxious and may induce neurodegenerative disorders, including Parkinson’s disease (PD) and Alzheimer’s disease (AD). The gut is the primary site for the overproduction of NO concentrations [57]. Previous studies revealed that gut lactobacteria, bifidobacteria, and *E. coli* can convert nitrate and nitrite into NO and can also stimulate host epithelial cells to form NO [58]. NO synthesis by gut microbiota can impact NO levels in the brain, and anomalous overproduction of NO leads to neuroinflammation due to its free radical properties, which compromise cellular integrity and viability and result in mitochondrial dysfunction through mitochondrial fission [56,59]. NO also excites a down-regulated secretion of brain-derived neurotrophic factor, which is essential for neuronal survival, development, and differentiation; synaptogenesis; and learning and memory [56]. NO can trigger amyloid beta deposition, and the accumulation of amyloid beta will compromise CNS functions [60].

## 3. Disturbance of Gut Microbiota Leads to Neurodegeneration

In recent years, there has been increasing evidence that intestinal microflora is associated with neurodegeneration to a certain extent. Alterations in the homeostasis of the microflora intestinal tract may worsen the etiology, pathogenesis, and/or progressions of some diseases, such as Multiple Sclerosis (MS), PD, AD, Amyotrophic Lateral Sclerosis (ALS), and other diseases [61]. It is well known that oxidative stress (OS) is closely related to mitochondrial dysfunction and is one of the main factors leading to neurodegeneration [62]. Interestingly, there are several pieces of proofs that microflora can interact with host cells by combining with mitochondrial activity [63]. Microflora imbalance will increase the level of reactive oxygen species (ROS) and magnify the OS scenario and neuronal inflammation [64]. Certainly, the diversity of intestinal microflora during aging is susceptible to accumulated dietary habits over many years, which may play a role in the development of neurodegeneration. It is well known that the composition of microbial community changes in diversity during aging [65], *Bacteriophyta* and *Phaeophyta* species are still dominant, although their ratio may change significantly. The increase in pathogenic bacteria is usually accompanied by a decrease in beneficial bacteria (symbiotic bacteria) [66]. One of the key factors in most neurodegenerative diseases is the formation of insoluble protein aggregates in neurons. Intestinal biological dysbiosis will give rise to the accumulation of toxic misfolded proteins in the conformation of β-Sheet, thus bringing about cell dysfunction, loss of synaptic connections, and neurodegenerative diseases [67].

AD is a common neurodegenerative disease with insidious onset and progressive course. The pathogenesis of AD is complicated, its pathological features mainly include abnormal deposition of extracellular amyloid protein (Aβ) outside nerve cells and formation of senile plaque (SP), as well as hyperphosphorylation of tau protein inside nerve cells to form neurofibrillary tangles (NFT) [68]. Aβ deposition can lead to neuronal damage, which is often considered to be the initial link in the pathogenesis of AD, and then a series of subsequent inflammatory reactions promote the progress of AD (Figure 3) [69]. Neuroinflammatory reactions have been found to cause apoptosis or necrosis of nerve cells, and eventually create irreversible damage to the brain. Studies have confirmed that compared with conventional AD mice (APP/PS1 mice), aseptic AD mice showed lower levels of Aβ, higher levels of Aβ dehydrolyzing enzyme, and less amyloidosis in the brain [70]. However, aseptic AD mice revealed worse pathological conditions like Aβdeposition after transplantation of bacteria from conventional AD mice. The composition of intestinal microorganisms in APP/PS1 mice was quite different from that in normal mice. Many intestinal microorganisms can secrete amyloid proteins themselves. Large amounts of amyloid proteins from bacteria and fungi such as CsgA and Aβ42 have certain neurotoxicity and can induce inflammation [71,72]. All these reminder intestinal microorganisms play an important role in the process of Aβ deposition.

Similarly, inflammatory response participates in the development of PD; many inflammatory factors in the brain of patients with PD, such as TNF-α and IL-6, are elevated [73]. Furthermore, the activation of microglia also plays an important role in the pathophysiological process of PD. Under the action of activated microglia cells and inflammatory cytokines, neurons are dysfunctional and necrotic [74]. Using α-synuclein overexpression mice, it was confirmed that intestinal microorganisms were involved in dyskinesia and activation of microglia. After administration of antibiotics, the dyskinesia was improved, the activation of microglia was alleviated, and the expression of TNF-α and IL-6 was decreased [75]. When the bacteria from PD patients were transplanted into α-synuclein overexpression mice, the related nerve injury would be aggravated. Interestingly, according to a study, the transplantation of fecal microflora or the transfer of fecal bacteria from healthy donors into the intestines of sick recipients might help rehabilitate and alleviate the symptoms of PD in human patients [76]. However, transplants of fecal samples from human PD patients into sterile mice enhanced the expression of αSyn and induced motor dysfunction, whereas humanized mice from matched healthy controls did not develop dyskinesia [77]. These findings suggest a direct relationship between intestinal microflora and the physiological symptoms of PD.

ALS is a fatal neurodegenerative disease characterized by the loss of upper and lower motoneurons as well as muscular atrophy. Even though there is a lack of clinical evidence for the involvement of intestinal microorganisms in the pathogenesis of ALS, intestinal leakage and blood–brain barrier disruption have been observed in animal models of amyotrophic lateral sclerosis [78,79]. For instance, ALS is associated with a reduced level of anti-inflammatory bacteria in the intestinal tract. Specifically, butyric acid-producing bacteria, including fiber isolates of *Vibrio butyricum*, *Escherichia coli*, *Oscillatory bacilli*, *Anaerobes*, and *Neisseria gonorrhoeae* are reduced [80]. The reduction of these butyric acid-producing bacteria correlated with the increased levels of pro-inflammatory cytokines in the gut and serum. Additionally, the ratio of *Bacteroides* spp. to mycelium in patients with ALS was disturbed compared with healthy people, providing further evidence for GM disorders in the pathology of ALS [81].

Taken together, these research outputs highlight the important link between intestinal biological disorders and neurodegenerative diseases and show the potential of intestinal microflora as a treatment or prevention option [82].

## 4. Mechanisms of the Neuroprotective Effect of TPs

### 4.1. Antioxidation

The human brain consumes about 20% of the oxygen inhaled, but its antioxidant activity is lower than that of other organs [83,84]. This raises the possibility of elevated ROS levels in the brain, which may have serious health implications, such as mitochondrial dysfunction and apoptosis, ultimately giving rise to neurodegenerative diseases like AD and PD [85]. A large number of studies have shown that TPs have strong antioxidant activity, which can protect the brain nerves.

#### 4.1.1. Directly Scavenge Active Oxygen Free Radicals and Nitrogen Free Radicals

The structure of TPs is rich in phenolic hydroxyl groups, and many researches have demonstrated that these chemical groups could act as free radical scavengers [86,87]. Indeed, the phenolic group is capable of donating either one hydrogen atom or a single electron to the ROS, stabilizing the reactive species and giving rise to a relatively stable flavonoid phenoxyl radical [87]. Through this mechanism, TPs can sacrificially react with free radicals such as hydroxyl (HO•), superoxide (O_2_•^−^), nitric oxide (NO•), alkoxyl, and peroxyl radicals [86].

#### 4.1.2. Inhibition to Oxidase Promotion and Lipid Peroxidation

In the rat model of cerebral ischemia and hypoxia, epigallocatechin gallate (EGCG) directly inhibited the expression of neuronal nitric oxide synthase (nNOS) gene, decreased the production of nitric oxide (NO), as well as reduced NO-induced oxidative damage [88]. nNOS is an enzyme deputed to the biosynthesis of NO, and overproduction of NO contributes to the pathogenesis and progression of neuronal diseases in different conditions of neuronal damages [89]. Meanwhile, TPs can prevent the increase of ROS and NO contents in rat midbrain and striatum induced by 6-hydroxydopamine [90]. EGCG, EGC, and ECG in catechin are thought to inhibit β-secretase activity, which can metabolize Aβ. Given this, EGCG has been found to inhibit Aβ-induced neurotoxicity, because it activates glycogen synthase kinase-3β (GSK-3β) and inhibits cytoplasmic non-receptor tyrosine kinase c-Abl/FE65, which is involved in nervous system development and nuclear translocation [91].

#### 4.1.3. Activate Intracellular Antioxidant Defense System

The intracellular antioxidant defense system mainly includes antioxidant enzymes, such as superoxide dismutase; catalase; glutathione S-transferase; and some low molecular compounds like vitamin C, vitamin E, and GSH, which can timely scavenge excess free radicals in the body to maintain the dynamic balance of free radicals [92]. EGCG was observed to markedly reduce mitochondrial oxidative damage through complex I saving, restoring ATP synthase activity, and increasing Sirt1-dependent PGC-1DNA deacetylation, T-FAM, NRF-1 levels, as well as mitochondrial alpha content in cell cultures of human lymphoblasts [93]. According to another study, molecules of EGCG directly inhibited the uptake of mpp^+^ or inhibited the transcription of iNOS (inducible nitric oxide synthase) by avoiding the binding of NFκB to the iNOS promoter. In the BV2 cell line (a murine microglial cell), EGCG pretreatment effectively inhibited Aβ-induced iNOS gene expression, blocked the formation of NO, and enhanced cellular antioxidant defenses [94].

### 4.2. The Activity of the Metal Chelating Agent

TPs are effective chelating agents for transition metals like iron and copper; this function is determined by the hydroxyl groups on the C-3 and C-4 positions of the B-ring, and the hydroxyl groups on the C-3 position of the C-ring, or determined by the three hydroxyl groups in some polyphenols [95]. For instance, EGCG is an important iron-chelating substance, the 3-dihydroxyl groups and gallic acid group in its B-ring may restore divalent iron ions to inactive iron atoms, thereby protecting cells from an oxidative stress injury. Apart from this, EGCG can inhibit more than 90% of DNA damage by chelating metal ions [96]. In addition, the pKa and IC_50_ values of polyphenols inhibiting the neurotoxicity of divalent iron ions and hydrogen peroxide indicated that the binding of polyphenols with iron ions was essential for their antioxidation [97]. It is well known that the level of oxidative stress in the brain region is closely related to the accumulation of divalent iron ions and the neurodegeneration of PD. Notably, the antioxidant and metal coordination properties of TPs were found to play an important role in the treatment of PD [98]. In this regard, the combined analysis showed that EGCG could inhibit the chemical reactivity of Fe^2+^ in the Fenton reaction by forming an NGAL–EGCG–iron complex, thus preventing the continuous aggravation of free radicals [99].

### 4.3. Regulate Signal Pathway

#### 4.3.1. Selectively Activate Protein kinase C (PKC) in Cerebral Neurons

PKC is a neuronal survival factor that participates in cell growth and differentiation and is involved in the formation and consolidation of various types of memory. The activation of PKC in neurons is the premise of playing some neuroprotective roles [100]. In this regard, the activation of PKC by EGCG protects SH-SY5Y cells and PC12 cells from cell death induced by 6-hydroxydopamine and Aβ [101]. It was found that after taking EGCG (2 mg/kg) for 2 weeks, PKCα in the striatum of mice was significantly upregulated [102], and PKCα in the hippocampal membrane and cytoplasmic part of mice was increased [103]. Meanwhile, EGCG could significantly increase the content of soluble amyloid precursor protein, which has anti-excitotoxicity and anti-oxidative damage induced by PKC [103]. Besides, EGCG has been observed to significantly restore the motor activity of the drosophila model expressing α-synuclein and reduce the apoptosis mediated by lipid peroxidation and oxidative stress [104].

#### 4.3.2. Regulate Other Signal Transduction Pathways

Except for PKC, other signaling pathways, such as mitogen-activated protein kinase (MAPK), 3-phospholipid phthalositol kinase (PI3K)/serine/threonine-protein kinase (Ser/Thr), AKT, and protein kinase A (PKA) signal cascade may also be involved in the neuroprotective effect of catechin. These cascades play an important role in protecting neurons from a variety of extracellular injuries and are also necessary for neuronal differentiation as well as survival [105]. Some studies have shown that catechins protected neurons from cell death caused by exogenous oxidative stress inducers through regulating the activity of MAPK. Additionally, these studies on the activation of Akt and ERK1/2 by EGCG suggested that the expression of antioxidant enzymes induced by EGCG may be accomplished by activating Nrf2 through Akt and ERK1/2 signaling pathways, which enable cells to acquire the ability of the antioxidant defense to survive under oxidative stress [106]. In a recent study, EGCG effectively inhibited sevoflurane-induced neurodegeneration and enhanced the memory and learning retention ability of mice via activating CREB/BDNF/TrkB-PI3K/Akt signal [107]. In PC12 cells, EGCG can also activate 22pGC-1α molecules through the SIRT-1 signaling pathway, consequently playing a neuroprotective effect against MPTP toxicity [108].

#### 4.3.3. Anti-Apoptosis

In SH-SY5Y neurons, the low concentration of EGCG could reduce the pre-apoptotic genes of bax, bad, caspase-1, and caspase-6. EGCG was also observed to play a neuroprotective role in oxidative excitation-mediated apoptosis of neuronal differentiated PC12 cell through downstream signaling of poly ADP-ribose polymerase and caspase aspartate proteinase (CASP)-3, or via upstreaming PI3K/Akt signal and GSK-3 cascades [109]. It was reported that the neurotoxin 6-hydroxydopamine (6-OHDA) was inhibited by EGCG by activating PKC phosphorylation [110]. According to the study of Levites et al. (2002), EGCG showed neuroprotective effects by restoring 6-OHDA-induced phosphorylation levels of ERK1/2 and PKC and by regulating anti-apoptosis related genes [111]. The neuroinflammation in PD incurs the expression of inflammatory mediators such as tumor necrosis factor-α and interleukin, which plays an important role in the initiation of apoptosis [99]. In the rat model of PD evoked by rotenone, EGCG treatment was responsible for having a neuroprotective effect on signally down-regulating the expression of tumor necrosis factor-α, interleukin-1, and interleukin-6 [112].

Astrosporin (STS) was observed to induce neurite collapse and apoptosis in isolated and cultured rat hippocampal neurons, while TPs could eliminate these adverse effects to a great extent and maintain the neuronal morphology [113]. STS can reduce the expression of Pro-BDNF, down-regulate the expression of the TrkB/Akt/Bcl-2 signal axis, and promote the activation of ERK1/2 and caspase-3. On the contrary, TPs can save the expression of pro-BDNF and recover the antagonistic biochemistry of the above signal effector molecules. Since the inhibition of TrkB or Akt by small molecular chemicals K252a and LY294002 can inhibit the activity of TPs, BDNF-TrkB, and Akt signaling, axes play a crucial role in TPs-mediated neuroprotection [114] (Figure 4). Overall, TPs have beneficial effects in protecting neurons from exogenous damage, such as STS-induced neurotoxicity and cell death. Moreover, EGCG also improved memory in PS2 mice by inhibiting ERK and NF- βB pathways to induce γ-secretase and β-secretase activity [115].

### 4.4. Regulate the Level of Neurotransmitters

Acetylcholine is an important neurotransmitter in the central nervous system. The choline system plays a special role in cognitive functions such as learning and memory [116]. Over the past several years, researchers have revealed that choline acetyltransferase (ChAT) activity decreased and acetylcholinesterase (acetylcholinesterase, AchE) activity increased in patients with AD, which led to a decrease in acetylcholine (Ach) level and occurrence of dementia symptoms mainly characterized by learning and memory impairment and cognitive impairment [117]. However, green TPs have a strong inhibitory effect on the activity of acetylcholine acetylase, suggesting it may be used to treat AD [118]. It was demonstrated that EGCG can up-regulate the level of nicotinic acetylcholine receptor (nAChR), inhibit the defect of nAChR induced by Aβ, and inhibit the decrease of cholinergic receptor neurons [119].

DA is a neurotransmitter that transmits signals needed for muscle movement coordination by regulating the excitability of striatal neurons under normal physiological conditions. The loss of dopamine will lead to irregular patterns of neural discharge and loss of motor control [120]. Nevertheless, it was found that Green TPs had an inhibitory effect on dopamine presynaptic carriers, which blocked the uptake of 1-methyl-4-phenyl-pyridine (MPP^+^) to protect dopaminergic neurons from MPP-induced damage [121]. Furthermore, EGCG inhibits the activity of catechol-O-methyltransferase (COMT) in rat hepatocyte fluid [122], dopamine, and related catecholamines, which are the substrates of COMT. Therefore, the suppression of EGCG increases the amount of dopamine in synapses. Neurotoxin 1-methyl-4-phenyl-1,2,3,6-tetrahydropyridine (MPTP), 6-OHDA was pointed out as selectively damaging the substantia nigra of the brain, thus leading to the loss of dopaminergic neurons [123]. Notably, the treatment of catechin extract improved the dyskinesia of crab-eating monkeys administered with MPTP, restored the levels of tyrosine hydroxylase (TH) and dopamine, decreased the level of the α-synuclein oligomer, as well as reduced their aggregation [124]. Even more interesting is tea extract can block the reduction of TH, an enzyme that catalyzes tyrosine to L-dihydroxyphenylalanine (L-DOPA) in the dopamine biosynthesis pathway, finally preventing the death of dopaminergic neurons [125]. It is known that Glutamic acid (Glu) is one of the most important endogenous excitatory neurotransmitters; its excessive release will produce neurotoxicity, which is closely related to dementia, depression, and other diseases [126]. In the organotypic culture of the motor neuronal spinal cord in rats, EGCG was observed to block the Glu excitotoxicity induced by threonine-hydroxyapatite (THA) and regulate the level of Glu in cells. In N18D3 cells (mouse heterozygous sensory neurons), EGCG inhibited quinolinic acid-induced NMDA receptor activation, and hence prevented apoptosis [127].

### 4.5. Autophagy

Autophagy is an internal process that helps lysosomes degrade and remove old and unnecessary cellular molecules, involving proteins, ribosomes, fat droplets, and other organelles, thereby maintaining the cell dynamic balance and cell survival under metabolic stress [128]. Accordingly, autophagy protects the overall health of the host, and the activation of autophagy in AD is conducive to neuroprotection as damage to autophagy may give rise to the accumulation of Aβ in the host [129]. Presently, TPs were observed to activate autophagy through different mechanisms, including the mammalian target of rapamycin pathway in HEK293T cells during endoplasmic reticulum stress and AMP-activated protein kinases [129]. As well, EGCG treatment induces autophagy because of attenuating negative regulators of autophagy, such as Gadd34, which controls apoptosis. In other words, EGCG prolongs autophagy by delaying apoptosis-mediated cell death and ultimately prolonging cell survival [130]. The optimal concentration of EGCG was proved to induce autophagy as well as an anti-inflammation effect, to degrade fat droplets in endothelial cells, and to promote the degradation of endotoxin, thus producing an anti-inflammatory effect [131,132].

## 5. The Reciprocal Interactions between TPs and Gut Microbiota Promote Nerve Protection

### 5.1. Biotransformation of TPs by Gut Microbiota

Most of the polyphenols in plants ingested from the diet are transformed into the intestine before being absorbed by intestinal and colon cells, resulting in the production of compounds with potential biological activity like low molecular weight metabolites [133]. The absorption and metabolism of TPs in the human body mainly depend on their biotransformation in the intestinal tract, so their bioactivity will be improved after transformation [134]. Most TPs are thought to be residual in the intestine and be converted into lactic acid type I metabolites (lactone, phenolic acid and aromatic acid, simple phenols) or type II metabolites (glucuronates, sulfates and oxy methyl derivatives), and then are converted into intermediate metabolites after enzymatic glycosylation by colonic bacteria, dehydroxylation, and demethylation by intestinal microorganisms [135]. When these metabolites are further transformed into small molecular compounds, they will enter the hepatointestinal circulation or systemic circulation and play a variety of physiological functions, which finally will be excreted through urine or feces. The metabolic transformation of TPs is not only modified by enzymes produced by hepatocytes and intestinal cells, but also by the enzymes produced through intestinal flora to reduce the potential toxicity of TPs, promote the formation of polyphenol metabolites, and show new biological activities [136].

Different intestinal bacteria were shown to degrade EGCG, such as *Enterobacter aerogenes*, *Escherichia pneumonia*, *Klebsiella pneumonia*, and *Bifidobacterium longum* selected from 169 strains of intestinal bacteria [136]. Using the porcine cecum model, it was found that EGCG was almost completely metabolized by porcine intestinal microflora within 4–8 h [137]. The degradation of EGCG by intestinal microflora involves a wide range of transformations, such as hydrolysis, C-ring cleavage, and reduction. It was reported that EGCG is first hydrolyzed to EGC and gallic acid by rat intestinal bacteria and bacterial strains. Intestinal flora in infants was also found to hydrolyze EGCG into EGC and gallic acid [138]. In addition, intestinal bacteria involved in the C-ring division of EGCG include redox bacteria, *Artoxicobacter popularism*, *Pseudomonas crematoria* or *Clostridium oxysporum* and other bacteria belonging to the genus *Vibrio butyricum* [139]. After the C-ring division, the remaining two free phenol rings underwent dehydroxylation. Notably, eubacteria strain SDG-2 can also degrade catechins through the division of the C-ring and the dihydroxylation of the B-ring [140]. Furthermore, it was suggested that intestinal bacteria can degrade EGCG to 5-(3-dihydroxy phenyl)-γ-enterolactone, and its glucuronic acid form is the primary urinary metabolite of EGCG [141]. Catechins are known to be degraded by microorganisms in the mouth and intestine; the involvement of microorganisms in catechin degradation was also demonstrated in humans and mice treated with antibiotics [142]. A review article also pointed out that most of the catechins ingested pass through the gut and undergo cyclic fission degradation [143]. The hydrolysis of the EGCG and ECG ester bonds and the cleavage of the C-ring are accomplished by microbial enzymes in the gut rather than mammalian enzymes. The bound TPP metabolites are degraded by microbial glucuronidase, demethylase, and sulfate esterase in the large intestine, and some metabolites (including catechins) can be reabsorbed and be further biotransformed [144]. It was reported that gallic acid has various biological activities among the biodegradable metabolites of TPs [145]. Others also have biological activities, but these activities are only shown in vitro and are usually lower than those of EGCG [146]. 3,4-dhpl, as the most abundant metabolite of catechins, down-regulates NF-jB transcription by phosphorylating IKK and IjB, and has stronger anti-adhesion and antioxidant activity than catechins. Other related metabolites of TPs have also shown anti-proliferative effects and anti-inflammatory activities [146], as well as anti-adhesion effects [147]. Moreover, the metabolites with tri- (or di-) hydroxyphenyl structure are expected to maintain their antioxidant activity. A large number of studies have shown that TPs metabolites derived from microorganisms had longer persistence in human circulation, so they could exert longer biological effects than their parent polyphenols [148].

Taken together, these results bring about the assumption that microbial metabolites not only enhance the absorptivity and bioavailability of maternal polyphenols but also bear primary responsibility for the potential to promote health at the organizational level.

### 5.2. Regulatory Effect of TPs on Intestinal Flora

Intestinal microflora has aroused great interest in recent years because of their potential impact on human health. The imbalance in the composition of intestinal microflora, known as biological disorders, is associated with obesity, type 2 diabetes, inflammatory bowel disease, and even PD [149,150]. After the tea is ingested by the host, TPs will be metabolized by microorganisms and will also affect the composition of microflora in the digestive tract [151].

It is estimated that TPs account for about 1/3 of the dry weight of tea; many studies have confirmed the regulatory effect of these components on intestinal microflora [152]. For example, it was proved that TPs increased the abundance of *Bifidobacterium* in C57BL/6 apolipoprotein E mice. We have evaluated the effect of (−)-epigallocatechin 3-*O*-(3-*O*-methyl) gallate (EGCG3”Me) intervention on the intestinal microflora of human flora-related model mice fed with a high-fat diet. The results showed that EGCG3”Me could significantly reduce the mycelial abundance and increase the abundance of *Bacteroides* and *Proteus* species [153]. According to a report, EGCG3”Me extracted from oolong tea could effectively improve the imbalance of intestinal flora in mice caused by a high-fat diet and significantly decrease the proportion of intestinal microflora in mice [154]. In another paper, Ikarashi et al. (2017) expressed that EGCG treatment reduced the level of *Clostridium* spp. [155]. Wang et al. (2018) researched the effects of green TPs on the colonic microflora of human flora-related C57BL/6J mice treated with HFD; the results revealed green TPs could effectively alleviate the decrease in microbial diversity induced by HFD [156]. Reversal of the HFD-induced Firmicutes/Bacteroides ratio also showed that supplementation of oolong TPs could significantly increase the diversity of fecal microflora and significantly reduce the fecal microorganism/*Bacteroides* ratio. At the same time, the relative abundance of *Bacillus* spp. increased, while that of *Clostridium* spp. and negative bacilli decreased significantly [157].

In addition to a single type of tea, many in vitro and animal studies have investigated the regulatory effects of a variety of teas on intestinal microflora. Sun et al. conducted an in vitro study to evaluate the regulatory effects of green, oolong, and black TPs on the human intestinal microflora [158]. The results showed that green tea, oolong tea, and black TPs could add the number of bifidobacteria in fermentation liquid, and the number of bifidobacteria increased gradually in the process of fermentation, among which the effect of oolong TPs was the most significant. Henning et al. (2018) evaluated the effects of decaffeinated green tea and black TPs on the cecal microflora of obese mice [159]. It was found that both green tea and black TPs could reduce the proportion of cecal *Bacteroides* spp. and increase the proportion of *Bacteroides* spp. Further analysis suggested that the regulatory effects of green and black TPs were different. In particular, green TPs increased the proportion of *Clostridium* and *Cocci* spp. and decreased the contents of *Turicibacter* and *Marvinbryantia* spp., while black TPs increased the proportion of *Oscillibacter*, *Anaerotruncus*, and *Pseudobutyrivibrio* spp. [160].

Flavanols, such as catechin, epicatechin, epicatechin gallate, and epigallocatechin gallate are considered to be the main dietary sources of flavanols due to widespread tea consumption in many countries [161]. Flavanols are the least glycosylated flavonoids, thereby having the highest bioavailability. Also, flavanols have different effects on bacterial growth in in vitro study of Manyin. Pure bacteria cultured with catechin and epicatechin inhibited the growth of intestinal pathogens *Staphylococcus aureus* and *Salmonella typhimurium* at concentrations of 125 g/mL and 1000 μg/mL, respectively [162]. Catechin and epicatechin inhibited the growth of symbiotic *Escherichia coli* and *Lactobacillus rhamnosus* at concentrations of 250, 500 or 1000 μg/mL. In contrast, when bacterial cultures were treated with lower concentrations of pure catechins (20,100 μg/mL), no inhibitory effect was observed against *Lactobacillus galactobacillus*, *Lactobacillus* spp., *Streptococcus* spp., *Colletobacter gonorrhea*, or *Escherichia coli* spp. [163].

In batch culture and fermentation studies with human fecal bacteria, tea flavanols (such as catechin and epicatechin) inhibited the growth of several pathogens such as *Clostridium difficile*, *Clostridium perfringens*, suppurative *Streptococcus* spp., and *Streptococcus pneumoniae* [164], along with having an inhibitory effect on the growth of symbiotic anaerobes such as *Clostridium.* However, probiotics such as *Bifidobacterium* and *Lactobacillus* spp. were not affected. In the same study, it was also observed that in batch culture, the concentrations of catechin and epicatechin decreased sharply after 24 h, while the concentrations of their microbial metabolites (such as 3-phenyl propionic acid and 4-hydroxyphenyl acetic acid) increased [165]. These microbial metabolites had little effect on symbiotic bacteria, whereas their parent compounds showed strong inhibitory effects on the growth of various pathogens, such as *Clostridium perfringens*, *Escherichia coli* O157:H7, *Pseudomonas aeruginosa*, *Salmonella enteritis*, *Salmonella typhimurium*, and *Staphylococcus aureus* [151].

## 6. Conclusions

In this review, the regulatory effect of TPs on intestinal flora and the protective effect of TPs on brain nerves were studied from the perspective of the microflora–brain–gut axis. Intestinal flora plays an important role in neurodegenerative diseases and affects the metabolic process and bioavailability of TPs in the body. Future research should further clarify the specific pathways of communication between intestinal microflora and brain through the mechanisms of neuroimmune, endocrine, and bacterial metabolites, and make a more in-depth study on the bioactive compounds with neuroprotective effects in TPs.

## Figures and Tables

**Figure 1 molecules-26-03692-f001:**
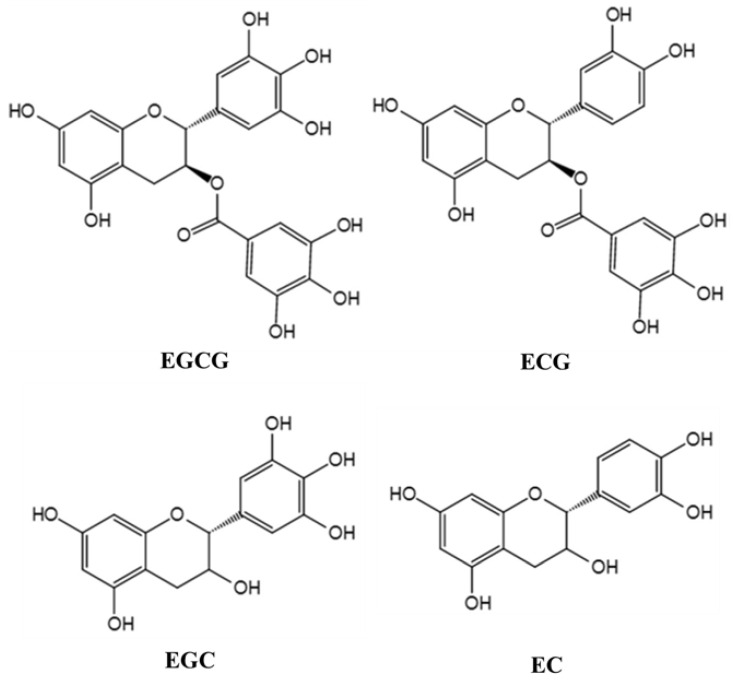
Molecular structure of the major TPs. The chemical conversion of intestinal microorganisms to biological heterologous matter, including food, industrial chemical pollutants, and drugs, directly affects their biological activity, toxicity, and efficacy. The metabolism and absorption of dietary polyphenols in the human body are also mainly dependent on the biotransformation of intestinal microorganisms [12]. Some of the polyphenols consumed by the human body directly enter the colon, the other part is converted into type I metabolites such as lactone, phenolic acid, and aromatic acid. Type II metabolites like glucuronide sulfate and oxy methyl derivatives then re-enter the colon through the small intestine. In the colon, polyphenols are glycosylated under the action of bacterial enzymes and then are dehydroxylated and demethylated by intestinal microorganisms into intermediate metabolites [13]. To take it further, these metabolites are converted into small molecular compounds, which enter the hepatointestinal circulation or systemic circulation to perform various physiological functions. TPs are converted into small molecular phenolic acids under the action of intestinal microorganisms and then are methylated, sulfated, sulfated, or ringed into the bloodstream [14]. Polyphenols have been regarded as the third largest regulator of intestinal health besides probiotics and probiotics. Plenty of researchers have found that plant polyphenols can maintain the homeostasis of the intestinal microenvironment [15], stimulate the growth of symbiotic and beneficial microbiota, and inhibit pathogenic strains [16]. For instance, studies have shown that TPs and their derivatives significantly inhibited the growth of some pathogenic bacteria such as *Clostridium perfringens* and *Bacteroides* spp., but had little effect on symbiotic anaerobes, such as *Bifidobacterium* spp. and *Lactobacillus* spp. [17]. Therefore, this paper reviews the interaction between intestinal flora and TPs, focusing on the regulatory mechanism of both in neuroprotection.

**Figure 2 molecules-26-03692-f002:**
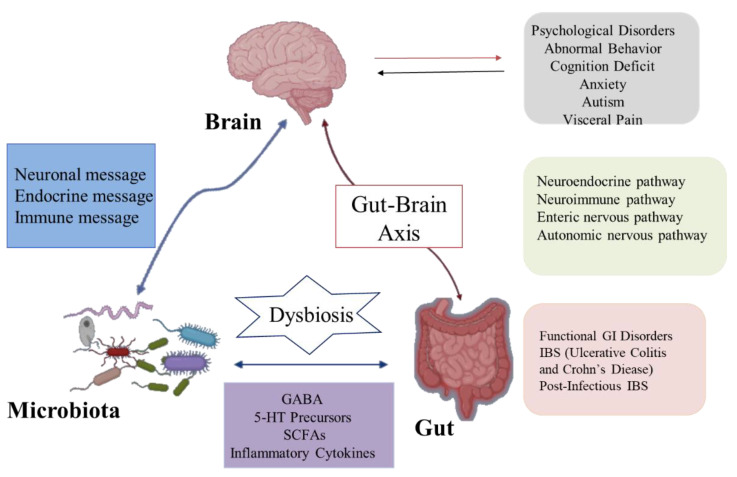
The gut microbiota communicates with the brain via the brain–gut axis.

**Figure 3 molecules-26-03692-f003:**
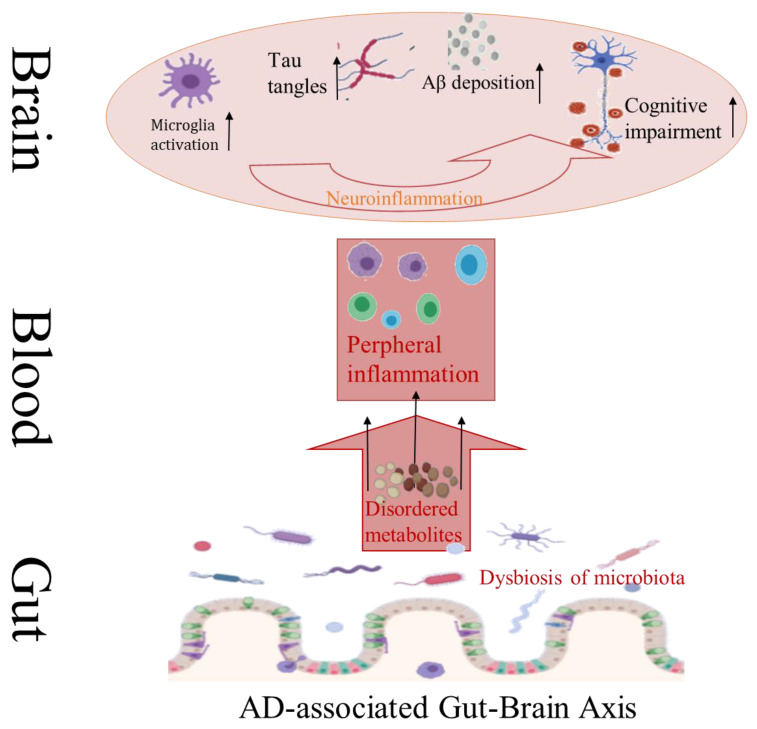
Major pathological features of AD associated with intestinal flora disturbance.

**Figure 4 molecules-26-03692-f004:**
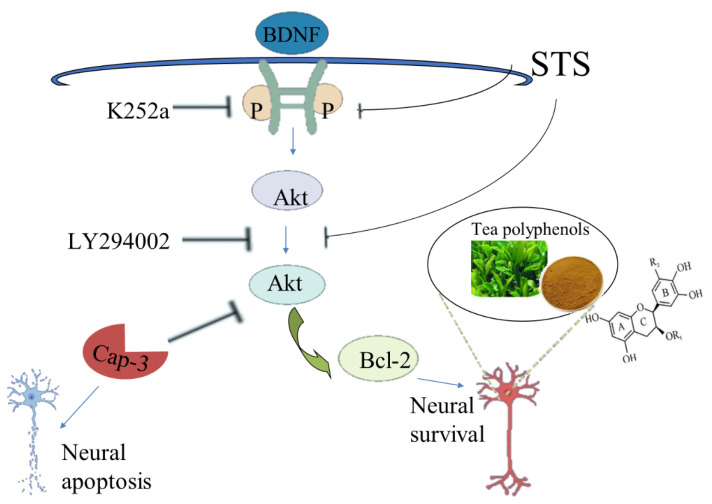
Tea polyphenols largely inhibit the collapse and apoptosis of neurite processes induced by astrosporins through the signal axes.

## Data Availability

No new data were created or analyzed in this study. Data sharing is not applicable to this article.

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
