# Peer review of "The Neuroprotective Effect of Tea Polyphenols on the Regulation of Intestinal Flora"

_molecules, 2021, doi:10.3390/molecules26123692_

Round 1
Reviewer 1 Report
The current manuscript reviews the interaction between intestinal flora and tea polyphenols, focusing on the regulatory mechanism of both in neuroprotection. In line 41 the first mention of the word "tea" full scientific name should be included in brackets.
The way in which scientific names are written is also very significant. To refer to members of a given genus in the plural sense, using Bacillus, Bifidobacterium, Lactobacillus, Anaerobes, Bacteroides, Clostridium, Proteus, Phaeophyta, Oscillibacter, Anaerotruncus, Pseudobutyrivibrio, Turicibacter and Marvinbryantia as examples, one cannot change the genus name directly to a plural form unless using sp. (one species) or spp. (more than one species)
In singular sense, it is suggested that they should be expressed as Bacillus sp., Bifidobacterium sp., Lactobacillus sp. , Anaerobes sp., Bacteroides sp., Clostridium sp., Proteus sp., Phaeophyta sp., Oscillibacter sp., Anaerotruncus sp., Pseudobutyrivibrio sp., Turicibacter sp. and Marvinbryantia sp.
In their plural form they should be addressed as Bacillus spp., Bifidobacterium spp., Lactobacillus spp. , Anaerobes spp., Bacteroides spp., Clostridium spp., Proteus spp., Phaeophyta spp., Oscillibacter spp., Anaerotruncus spp., Pseudobutyrivibrio spp., Turicibacter spp. and Marvinbryantia spp.

Author Response
Point 1: The current manuscript reviews the interaction between intestinal flora and tea polyphenols, focusing on the regulatory mechanism of both in neuroprotection. In line 41 the first mention of the word "tea" full scientific name should be included in brackets.
Response 1: The word "tea" full scientific name is Camellia sinensis (L.) O. Kuntze, and has been added in brackets in the revised manuscript.
Point 2: The way in which scientific names are written is also very significant. To refer to members of a given genus in the plural sense, using Bacillus, Bifidobacterium, Lactobacillus, Anaerobes, Bacteroides, Clostridium, Proteus, Phaeophyta, Oscillibacter, Anaerotruncus, Pseudobutyrivibrio, Turicibacter and Marvinbryantia as examples, one cannot change the genus name directly to a plural form unless using sp. (one species) or spp. (more than one species).
In singular sense, it is suggested that they should be expressed as Bacillus sp., Bifidobacterium sp., Lactobacillus sp., Anaerobe sp., Bacteroides sp., Clostridium sp., Proteus sp., Phaeophyta sp., Oscillibacter sp., Anaerotruncus sp., Pseudobutyrivibrio sp., Turicibacter sp. and Marvinbryantia sp.
In their plural form, they should be addressed as Bacillus spp., Bifidobacterium spp., Lactobacillus spp., Anaerobes spp., Bacteroides spp., Clostridium spp., Proteus spp., Phaeophyta spp., Oscillibacter spp., Anaerotruncus spp., Pseudobutyrivibrio spp., Turicibacterspp. and Marvinbryantia spp.
Response 2: Thanks for the professional and friendly suggestions. The scientific names in the manuscript have been checked and revised as you have advised. Please see the attachment of the revised manuscript.

Reviewer 2 Report
The topic of the present review is very interesting The authors have described the connections between a dysfunctional gut microbiota and the development of several neurological diseases. Then, the positive effects that the main constituent of the green tea, i.e. epigallocatechin gallate, has demonstrated in these diseases by means of different mechanisms occurring in the brain are presented. Finally, authors discuss the role of EGCG in the regulation of the intestinal flora, suggesting a potential connection of the therapeutic effects of EGCG in the gut microbiota-brain axis to prevent the development of neurological diseases. In my opinion the present manuscript can be accepted for publication after the following minor revisions:
In the introduction section, a figure of the molecular structure of the cited TPs should be added
The role of nitric oxide and Nitric Oxide Synthases in the cross talk between gut and brain should be discussed in more details. Please see: ACS Chem. Neurosci. 2017, 8, 7, 1438–1447
More chemical details on the ROS scavenging by TPs should be added in paragraph 4.1.1. A chemical scheme is highly suggested. In this regard, the cited reference 81 is not appropriate since it is not specifically related to the discussed mechanism and should be substituted
In paragraph 4.1.2 authors should refer to the specific specifical role that neuronal NOS has in the development of neurological disorders. Please see: Neural Regeneration Research 2016, 11(11), pp. 1731-1734
On p.8: line 333, please explain the acronym iNOS; in lines 318-319 check the names of the cited compounds
On p.12, lines 511-513: rewrite this sentence, avoiding to start with the grammatical conjunction “And”
On p.13, lines 540-541: the affirmation “EGCG is one of the most bioactive components in TPs” can be avoided since it was explained in the introduction section
Author Response
Point 1: In the introduction section, a figure of the molecular structure of the cited TPs should be added.
Response 1: A figure of the molecular structure of the cited TPs has been added in the introduction section in the revised manuscript (as Figure 1). Please see the attachment.
Point 2: The role of nitric oxide and Nitric Oxide Synthases in the cross talk between gut and brain should be discussed in more details. Please see: ACS Chem. Neurosci. 2017, 8, 7, 1438–1447.
Response 2: The role of NO and NO synthases in the cross talk between gut and brain has been discussed in more detailed in the revised version. The main discussion is as follows.
Nitric oxide (NO) acts as the principal neurotransmitter of the nonadrenergic noncholinergic ENS and performs a vital role in the Central Nervous System of humans and animals. However, excessive NO production becomes noxious and may induce neurodegenerative disorders, including AD and PD. The gut is the primary site for the overproduction of NO concentrations. Previous studies have revealed that gut lactobacteria, bifidobacteria, and E. coli Nissle 1917 can convert nitrate and nitrite into NO and can also stimulate host epithelial cells to form NO. NO synthesis by gut microbiota can impact NO levels in the brain, and anomalous overproduction of NO leads to neuroinflammation due to its free radical properties, which compromise cellular integrity and viability and result in mitochondrial dysfunction through mitochondrial fission. NO also excites a down-regulated secretion of brain-derived neurotrophic factor, which is essential for neuronal survival, development and differentiation, synaptogenesis, and learning and memory. NO can trigger amyloid beta deposition and the accumulation of amyloid beta will compromise CNS functions.
Point 3: More chemical details on the ROS scavenging by TPs should be added in paragraph 4.1.1. A chemical scheme is highly suggested. In this regard, the cited reference 81 is not appropriate since it is not specifically related to the discussed mechanism and should be substituted.
Response 3: More chemical details on the ROS scavenging by TPs have been added in the revised manuscript. The relating cited references have been substituted.
The structure of TPs is rich in phenolic hydroxyl groups, and many researches have demonstrated the polyphenols with the structure may exert antioxidative effects as free radical scavengers. The ROS scavenging-mediated antioxidant activity of polyphenols is primarily attributed to the presence of those benzene ring-bound hydroxyl groups that are capable of donating either one hydrogen atom or a single electron to the ROS, stabilizing the reactive species and giving rise to a relatively stable flavonoid phenoxyl radical. The flavonoid phenoxyl radical may react with a second radical (RO•), acquiring a stable quinone structure. Through this mechanism, TPs can sacrificially react with free radicals such as hydroxyl (HO•), superoxide (O2•-), nitric oxide (NO•), alkoxyl and peroxyl radicals.
Point 4: In paragraph 4.1.2 authors should refer to the specific specifical role that neuronal NOS has in the development of neurological disorders. Please see: Neural Regeneration Research 2016, 11(11), pp. 1731-1734.
Response 4: Specifical role of NOS in the development of neurological disorders has been emphasized, expressing as “nNOS is an enzyme deputed to the biosynthesis of NO, and overproduction of NO contributes to the pathogenesis and progression of neuronal diseases in different conditions of neuronal damages”.
Point 5: On p.8: line 333, please explain the acronym iNOS.
Response 5: iNOS is short for “inducible nitric oxide synthase”. The explanation has been added in the bracket.
Point 6: On p.8:in lines 318-319 check the names of the cited compounds.
Response 6: The names of the cited compounds in the sentences were checked and revised in the manuscript.
Point 7: On p.12, lines 511-513: rewrite this sentence, avoiding to start with the grammatical conjunction “And”
Response 7: The term “And that” can be deleted from the sentence. The whole paragraph is still complete and fluent.
Point 8: On p.13, lines 540-541: the affirmation “EGCG is one of the most bioactive components in TPs” can be avoided since it was explained in the introduction section.
Response 8: The affirmation “EGCG is one of the most bioactive components in TPs” has been deleted in the paragraph.
Round 2
Reviewer 2 Report
Authors have improved their manuscript according to my suggestions. I have found different typos and mistakes, therefore a careful language revision is necessary before publication. In particular:
p.7, line 231 eliminate the words "Nissle 1917 "
p.9 lines 333-339: the sentences should be rewrited as follows:
"The structure of TPs is rich in phenolic hydroxyl groups, and many researches have demonstrated that these chemical groups could act as free radical scavengers [86, 87]. Indeed, the phenolic group is capable of donating either one hydrogen atom or a single electron to the ROS, stabilizing the reactive species and giving rise to a relatively stable flavonoid phenoxyl radical [87].
Author Response
Point 1: Authors have improved their manuscript according to my suggestions. I have found different typos and mistakes, therefore a careful language revision is necessary before publication. In particular:
p.7, line 231 eliminate the words "Nissle 1917 "
p.9 lines 333-339: the sentences should be rewrited as follows:
"The structure of TPs is rich in phenolic hydroxyl groups, and many researches have demonstrated that these chemical groups could act as free radical scavengers [86, 87]. Indeed, the phenolic group is capable of donating either one hydrogen atom or a single electron to the ROS, stabilizing the reactive species and giving rise to a relatively stable flavonoid phenoxyl radical [87].
Response 1: Thanks for the professional and friendly suggestions. The typos and mistakes in the manuscript have been revised as you have advised. A careful language check has been carried out and details of modifications were marked up using the “Track Changes” in the MS Word version of the manuscript. Please see the attachment of the revised manuscript.